# The roles of feedback loops in the *Caenorhabditis elegans* rhythmic forward locomotion

Peng Zhao[1☯], Boyang Wang[2☯], Yi Rong[3☯], Ye Yuan[4], Jian Liu[4], Hong Huo[1], Zhuyong Liu[2]*, Zhaoyu Li[3]*, Tao Fang[1]*

**1** Department of Automation, Shanghai Jiao Tong University, Shanghai, China, **2** School of Naval Architecture, Ocean and Civil Engineering, Shanghai Jiao Tong University, Shanghai, China, **3** Queensland Brain Institute, The University of Queensland, Brisbane Queensland, Australia, **4** Institute of Machine Intelligence, University of Shanghai for Science and Technology, Shanghai, China

☯ These authors contributed equally to this work
* zhuyongliu@sjtu.edu.cn (ZL); zhaoyu.li@uq.edu.au (ZL); tfang@sjtu.edu.cn (TF)

## Abstract

Rhythmic behaviors are essential in biological systems, particularly in animal loco-motion. The central pattern generator and sensory feedback loop mechanism have been instrumental in explaining many rhythmic locomotion patterns, however, it is insufficient to account for the tunability and robustness of frequency and amplitude in certain oscillatory movements. This suggests the involvement of additional, less understood circuit mechanisms. This study employs calcium imaging and neurome-chanical modelling to investigate the circuit mechanism responsible for sinusoidal forward locomotion in *Caenorhabditis elegans*. We demonstrate that the feedback loop circuit, consisting of motoneurons and muscles, could govern the generation of oscillations and regulate rhythmic forward movement. This circuit is composed of both negative and positive feedback loops, which together regulate the turnability and robustness of oscillations. The oscillatory behavior of *C. elegans* typically involves a rhythmic alternation of dorsoventral muscles. Our neuromechanical model of the functional oscillatory unit reveals that asymmetric inputs from interneurons to moto-neurons, and asymmetric connections from motoneurons to muscles, are essential for this switching mechanism. Our findings suggest that, besides the established roles of existed oscillator mechanisms, circuits formed by both negative and positive feedback loops contribute to the generation and robust modulation of rhythmic behaviors.

## Author summary

The intricate circuitry underlying sinusoidal forward locomotion in *Caenorhabditis elegans* sheds light on the nuanced mechanisms guiding oscillatory movements. By uncovering the feedback loop circuit's impact on rhythmic behavior,

**Data availability statement:** All data and code for this article have been deposited on Zenodo at https://doi.org/10.5281/zenodo.12746051.

**Funding:** This research was supported by the National Natural Science Foundation of China (41571402 to Tao Fang, 12272222 to Zhuyong Liu), the Science Fund for Creative Research Groups of the National Natural Science Foundation of China (61221003 to Tao Fang), and Li lab startup funding to Zhaoyu Li. The funders had no role in study design, data collection and analysis, decision to publish, or preparation of the manuscript.

**Competing interests:** The authors have declared that no competing interests exist.

the research underscores the crucial role of both negative and positive feedback loops in regulating oscillatory behaviors. These findings not only enhance our comprehension of rhythmic locomotion but also suggest a broader perspective on how coupled mechanisms contribute to the generation and modulation of rhythmic behaviors.

## Introduction

Rhythmic behaviors, including locomotion, respiration, and feeding, are vital for animal survival and reproduction [1–7]. The oscillatory mechanisms underlying these rhythmic behaviors currently focus on two possibilities: central pattern generators (CPGs) and sensory feedback loops. Intrinsic oscillatory neurons or networks serve as CPGs and are pivotal in generating stereotyped rhythmic behaviors, such as flying [8], swimming [9], walking [10], running [11], and reproduction [12], in both vertebrates and invertebrates. For example, a spinal CPG with a positive input facilitates spontaneous rhythmic locomotion in zebrafish [9]. Notably, sensory feedback loops can also generate and modulate rhythmic behaviors across various conditions [13–15]. For instance, feedback signals contribute to the stabilization and control of the neuromuscular system during insect locomotion [13]. Nevertheless, the extent to which the oscillation generation and modulation are mediated by CPGs or alternative sensory feedback loop mechanisms remains incompletely understood.

*Caenorhabditis elegans* provides a promising model organism for investigating this question. *C. elegans* possesses a straightforward nervous system comprising merely 302 neurons and 95 muscles, facilitating research on circuits and locomotory behaviors. Previous studies suggest that the CPGs and feedback loops may both play roles in the regulation of oscillatory behaviors. For example, the head CPGs and those made up of motoneurons in the ventral nerve cord control muscle contractions that enable the worm to move forward or backward [16–20]. Simultaneously, sensory feedback loops involving motoneurons, muscles, and stretch receptors could generate rhythmic locomotion [21–23]. To elucidate the control circuit in greater detail, spring-rod models have been proposed in these studies. These models effectively capture the postures of locomoting worms. Nonetheless, the coupling between modules in traditional Newtonian mechanics models makes it difficult to accurately replicate oscillatory behaviors seen in real worms. Furthermore, recent studies have shown a positive relationship between motoneuron activity and body curvature [24,25], suggesting the presence of positive feedback loops at work. Moreover, mechanosensitive signals differentially modulates the typical rhythmic behaviors: swimming and crawling [26]. Nevertheless, the mechanisms by which rhythmic forward movement is generated and adjusted through negative and positive feedback loops remain poorly understood. Additionally, the neuromechanical model needs to address the issue of strong coupling to accurately describe this oscillatory process.

Here we employ the experimental calcium imaging and the theoretical neuromechanical model to elucidate the sensory feedback mechanism underlying oscillation

generation and to delineate the roles of negative and positive feedback loops in the forward locomotion of *C. elegans*. We demonstrate that the feedback loops, comprising motoneurons and muscles, are pivotal in governing oscillation generation and maintaining rhythmic forward movement. Our neuromechanical model of the functional oscillatory unit further clarifies that asymmetry is the critical condition for dorsoventral muscle switching during forward rhythmic locomotion. Our findings suggest that, in addition to the view of CPGs, functional circuits incorporating both negative and positive feedback loops can generate and modulate rhythmic behaviors, offering novel insights into oscillatory biological processes involving competitive dynamics.

## Results

### The neuromechanical model is built for *C. elegans* rhythmic behaviors

For the precise computation and analysis of the intricate, highly coupled oscillatory dynamics at the circuit level, a neuro-mechanical model is indispensable. Given the well-characterized neural and muscular dynamics [27,28], the fully mapped connectome [29–31], and quantitatively characterized oscillatory behaviors [32,33], *C. elegans* is an ideal model organism for constructing such a neuromechanical model. The oscillatory behavior in *C. elegans* involves the precise coordination of different muscles. Previous models based on Newtonian mechanics are insufficient in accurately modelling this process. Consequently, we adopt multibody mechanics, which is particularly suited for studying the locomotion mechanisms of systems comprising multiple flexible and rigid components. By integrating multibody mechanics with the *C. elegans* connectome and muscle dynamics, our objective is to construct a more accurate neuromechanical model.

We initially constructed a multibody system to simulate the coupled dynamics of the *C. elegans* body, based on the arrangement of its muscles. The body is modelled as 13 T-shaped rigid rods connected by 12 pairs of muscles (Fig 1A), which mimics the four quadrants and 95 muscle cells of the known musculature [34]. To streamline the representation of muscle function, the entire body was segmented into 12 equal segments (see Materials and Methods and S1Table).

Subsequently, we isolated a compact neural circuit responsible for forward locomotion from the *C. elegans* connectome. The RIB-SMD and AVB-B subcircuits are the two primary circuits driving forward locomotion [35–38]. The head-neck muscles, comprising the 1st to 3rd pairs of muscles, are primarily regulated by the RIB-SMD subcircuit, whereas the remaining 4th to 12th pairs of muscles are controlled by the AVB-B subcircuit. The SMDV, RMDV, and VB neurons govern the ventral muscles, while the SMDD, RMDD, and DB neurons control the dorsal muscles. The transition between ventral and dorsal muscle contractions is coordinated by the inhibitory motoneurons RMEV, RMED, VD, and DD. The multibody elements and interconnected neurons are depicted in part within the intricate neuromechanical diagram (Fig 1B). Our model accounts for the complex interactions between muscles and motoneurons, including sensory feedback from muscles to motoneurons, as indicated by previous research. Formally, $N$ membrane potential elements $\mathbf{x} = [x_1, x_2, \ldots, x_N]^\mathsf{T}$ and $M$ multibody mechanics elements $\mathbf{y} = [y_1, y_2, \ldots, y_M]^\mathsf{T}$ make up the dynamics of the neuromechanical model that follows the coupled nonlinear equations.

$$\begin{cases} \frac{d\mathbf{x}}{dt} = \mathbf{f}_1(\mathbf{x}, \mathbf{y}) \\ \frac{d\mathbf{y}}{dt} = \mathbf{f}_2(\mathbf{x}, \mathbf{y}) \end{cases} \tag{1}$$

Terms on the right-hand side of Eq 1 describe the dynamics of each component, where $\mathbf{f}_1(*) = [f_{11}(*), f_{12}(*), \ldots, f_{1N}(*)]^\mathsf{T}$ and $\mathbf{f}_2(*) = [f_{21}(*), f_{22}(*), \ldots, f_{2M}(*)]^\mathsf{T}$ capture the dynamics of membrane activities and multibody system, respectively. Feedback loops exist in $\mathbf{f}_1(*)$, described as feedback currents from muscles to neurons (Fig 1C). Muscle-driven forces, which transform membrane potentials into forces, and Muscles' passive forces, which express elastic and damping forces, come from $\mathbf{f}_2(*)$ (see Materials and Methods).

Following this, we refined the remaining unaccounted parameters of the neuromechanical model through the application of an evolutionary algorithm (EA) to elucidate rhythmic forward locomotion at the circuit level. The methodology for the

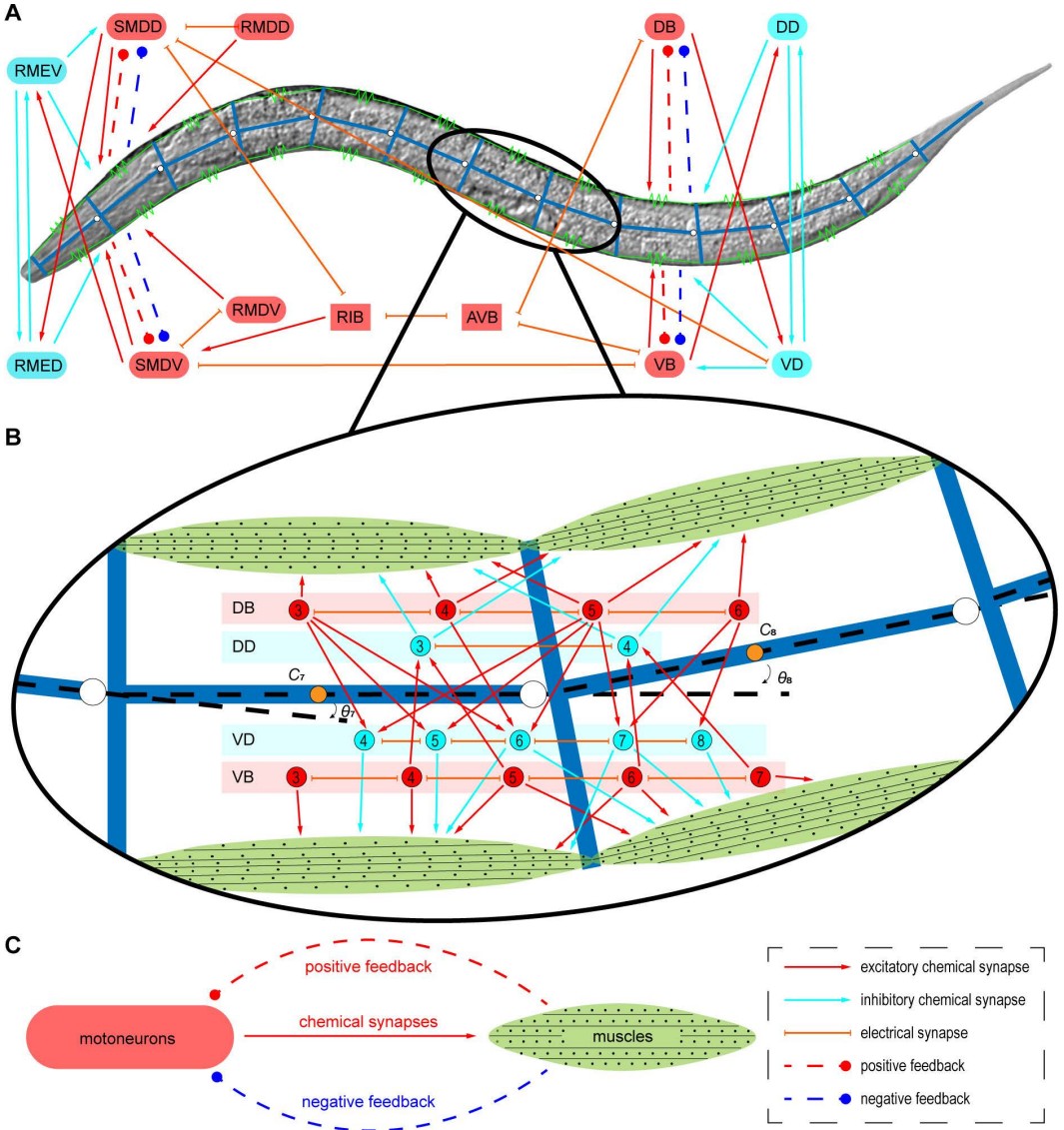

**Fig 1. The neuromechanical model for *C. elegans*.** (A) The neuromechanical model consists of the multibody system and neural circuits. The worm is made up of 13 rigid, T-shaped rods, joined by 12 pairs of damped springs. The RIB-SMD subcircuit mainly drives the first three units. Units from 4th to 12th are driven by AVB-B subcircuit. According to connectome data, circuits consist of gap junctions and chemical synapses. Sensory feedback from muscles to motoneurons is included. (B) The connections in the 7th and 8th units are shown in detail. The gap junctions between motoneurons are lines with endpoints. The arrow lines represent chemical synapses from motoneurons to muscles. Multibody notes include centers and angles. (C) Muscles and motoneurons make up feedback loops. Motoneurons control muscles through chemical synapses, while muscles influence motoneurons through positive and negative feedback, shown by dashed lines.

EA is detailed in the Materials and Methods section, and certain invariant parameters [27,28,39] of neurons and muscles are outlined in S2 Table. To enhance the modulation of amplitude and frequency, we subjected *C. elegans* to various Halo oils with differing viscosities. *C. elegans* exhibited oscillatory behavior in two dimensions, characterized by frequencies ranging from 0.5 to 1.3 Hz and velocities from 60 to 195 um/s when immersed in the mixed Halo oil (S1 Fig). These oscillatory characteristics were employed to optimize the unidentified parameters.

Consequently, through the utilization of this neuromechanical model, either a simple single feedback loop, a dorsoventral switching functional unit, or a coupled simulated complete worm could be analyzed for the purpose of investigating the locomotion mechanism of *C. elegans*.

## The oscillators are composed of negative neuron-muscle feedback loops

In the developed neuromechanical model, we proposed a feedback circuit. To delve into the intricacies of the feedback mechanism, we focused on identifying the source of the negative feedback. Recognizing that B-type motoneurons, or their associated neuronal networks, can produce oscillatory forward locomotion, in parallel, proprioceptive SMD modulates head muscles during complex gaits, such as oscillatory patterns [24,40], we investigated whether an artificially constrained worm would display regular oscillations within these neurons. Employing fluorescence microscopy, we observed and measured the calcium activities of B-type motoneurons during ventral and dorsal bending movements in the worms. To ascertain whether the oscillations were generated by circuits composed exclusively of motoneurons, we focused our analysis on the nearby motoneurons VB7 and DB5. Upon bending towards the ventral or dorsal side, the calcium activities of VB7 and DB5 nearly synchronized, but without periodic oscillations (Figs 2A, 2B, and S2). Conversely, during forward locomotion, the calcium activities of the adjacent DB and VB motoneurons must oscillate in the antiphase [25]. Thus, the oscillations of B-type motoneurons or networks of related motoneurons are unlikely to be self-oscillatory but may instead be induced by feedback from the muscles.

To ascertain the origin of the feedback-driving oscillations, whether they arise from muscle membrane potentials, we recorded the calcium activities of B-type motoneurons in worms rendered immobile under various light stimuli. It is established that action potentials drive the deformation of muscles in *C. elegans* [28]. Activation of muscles via optogenetic means elicits elevated calcium activities within these cells. Even when the muscles are paralyzed, they continue to 'activate' in response to optogenetic stimulation, although their shape remains unchanged. Our findings indicate that this activation does not inhibit the calcium activities of VB and DB motoneurons (Fig 2C), implying that the feedback is more likely due to muscle deformation rather than membrane potentials. These results indicate that negative feedback regulation may play a crucial role in oscillatory behavior from head to tail.

To comprehend the function of the negative feedback loop from the perspective of closed loops [41,42], we constructed a two-node negative feedback loop circuit composed of one muscle and one motoneuron (Fig 2D). In the model, a single B-type motoneuron or SMD motoneuron governs only the unilateral muscle of *C. elegans*. Negative feedback emerges because muscle deformation exerts a negative influence on the motoneuron. The EA calibrates unknown parameters to the forward locomotion with a frequency of 1 Hz (Fig 2E). Muscle deformation is oscillatory, as are the membrane potentials of motoneurons and muscles (Fig 2F). Elevated levels of negative feedback, the weight of the motoneuron relative to the muscle, and the muscle's contractile capacity result in an increase in frequency and amplitude (S3 Fig).

Therefore, our findings from the experimental and theoretical study, which combines fluorescence microscopy and the neuromechanical model, suggest that negative neuron-muscle feedback loops have the ability to drive the oscillatory forward locomotion of *C. elegans*. Thus, our neuromechanical model encompassing the feedback from muscle to neurons is rational.

## Coordination of oscillatory behavior requires dorsoventral asymmetry

The oscillatory behavior of *C. elegans* is dependent on the coordinated activity of both dorsal and ventral circuits. To accurately capture the ventral and dorsal coordination within our neuromechanical model, we developed a functional unit that incorporates a segment of the circuit. To differentiate the conditions for the switching of dorsoventral contractions, we employed two two-node negative feedback loops to create a competitive structure (Fig 3A). These feedback loops incorporate the positive signals transmitted from the DB/VB neurons to the muscles via chemical synapses, as well as negative feedback from the muscles back to the DB/VB neurons. The DD/RME neurons are responsible for muscle relaxation

PLOS Computational Biology

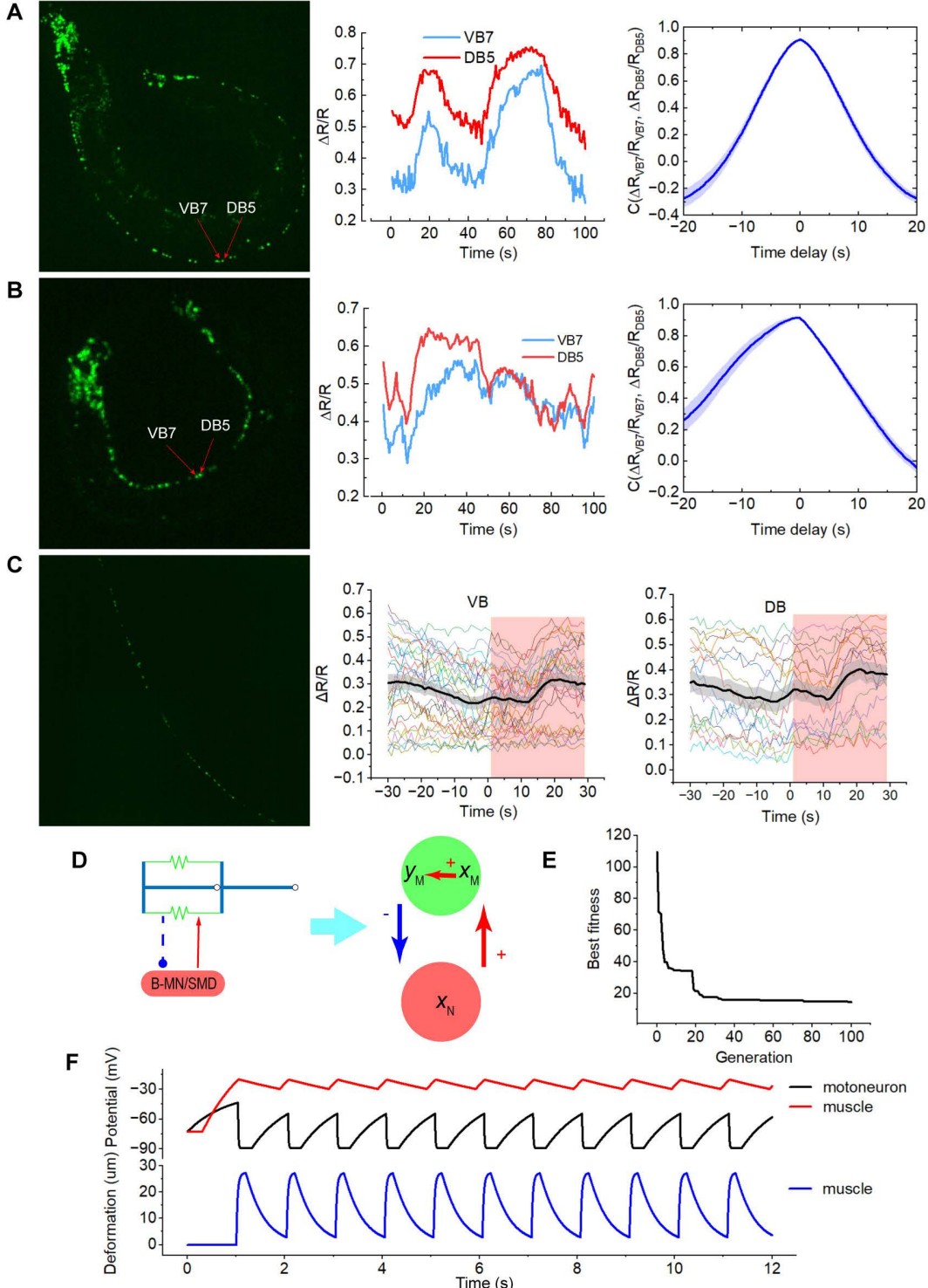

**Fig 2. Calcium activities of motoneurons and the two-node negative feedback loop to generate oscillation.** (A) Worms bent to the dorsal are seen on the left. The calcium activities of VB7 and DB5 are shown in the middle, with their cross-correlation on the right. Error bars mean SEM (n = 10). (B) Worms bent to the ventral are on the left. The calcium activities of VB7 and DB5 are shown in the middle, with their cross-correlation shown on the right. Error bars are SEM (n = 10). (C) On the left is the paralyzed worm with green, fluorescent neurons. The calcium activities of all VB and DB

motoneurons are shown in the middle and right, respectively. Error bars represent SEM (n = 29 and n = 17). (D) One muscle and one motoneuron (SMD or B-type motoneuron) make up the negative feedback loop. (**E**) the optimization process of the EA. (F) Membrane potentials of motoneuron, and muscle, muscle deformation are in oscillatory styles.

following contraction. The premotor neurons AVB and RIB serve as triggers for the initiation of forward oscillatory behavior. The optimal values for the unknown parameters within this functional unit were determined through EA optimization (S4 Fig).

Within this functional unit, oscillatory activity can be modulated by inputs from premotor neurons, chemical synapses between motoneurons and muscles, and negative feedback from muscles to motoneurons. Initially, we investigated the effect of premotor neuron inputs on oscillation. Since these inputs are connected to both the dorsal and ventral sides, we tested whether oscillation depends on asymmetric inputs to both sides. To this end, we adjusted the connection weights of the ventral side to equal those of the dorsal side, based on the optimized parameters. The results demonstrate that oscillation is nearly absent under symmetric conditions (Fig 3B). On the contrary, when the premotor neuron inputs are asymmetric, the muscle deformation exhibits oscillatory behavior (Fig 3B). This suggests that the asymmetric input might play a role in dorsoventral switching.

Subsequently, we embarked on a series of experiments to evaluate the influence of chemical synapses on oscillatory patterns. The DD/RME and DB/VB neurons establish synapses with the dorsal and ventral muscles, respectively. To ascertain the nature of these synapses, whether symmetric or asymmetric, we manipulated the weights of dorsal side connections while maintaining the constancy of the ventral side. Our results indicate that asymmetric synapses in the muscles are indispensable for the generation of oscillations (Figs 3C and S4). However, the observed asymmetry does not arise

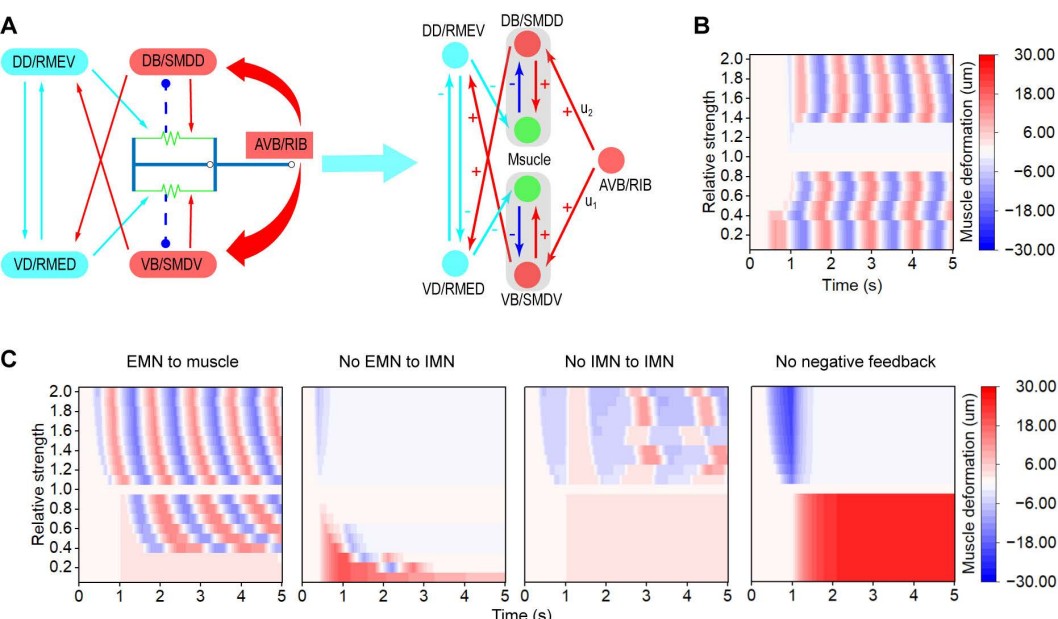

**Fig 3. The functional unit for dorsoventral switching.** (A) The dorsoventral switching model is made up of inhibitory motoneurons and two negative feedback loops. (B) Heatmap of muscle deformation. The vertical axis means inputs are different. Note that the relative strength of input is calculated by $u_2/u_1$. (C) Heatmap of muscle deformation. The weights of synapses from excitatory motoneurons (EMN) to muscles are different between the dorsal and ventral sides. Furthermore, three types of connections, synapses from excitatory to inhibitory motoneurons (IMN), synapses between inhibitory motoneurons, and negative feedback are ablated, respectively. Relative connection strength is calculated by $w_d/w_v$. $w_v$ is the optimized parameter at ventral side.

from excitatory to inhibitory motoneurons (S4 Fig) or from synapses between inhibitory motoneurons (S4 Fig). Recognizing that these two types of connections might serve additional functions, we conducted ablation experiments. We configured the chemical synapses from excitatory motoneurons to muscles to be asymmetric. Then, we ablated the synapses from excitatory to inhibitory motoneurons, which led to the disappearance of oscillations (Fig 3C). We then proceeded to ablate the synapses between inhibitory motoneurons, resulting in a decrease in oscillation frequency compared to when these synapses were present (Fig 3C). Additionally, we set the chemical synapses from inhibitory motoneurons to muscles to be asymmetric, and the ablation of the synapses between inhibitory motoneurons reduced the frequency of oscillation (S4 Fig). Therefore, the asymmetry from motoneurons to muscles also emerges as a critical factor in the coordination of dorsoventral switching.

Next, we sought to determine the role of negative feedback in dorsoventral switching. We applied negative feedback with varying weights on the dorsal side while keeping all other connections unaltered. Our results suggest that the symmetry or asymmetry of negative feedback does not affect the switching process (S4 Fig). To assess the necessity of negative feedback for the generation of oscillations within the functional unit, we selectively ablated this signal in the context of asymmetric chemical synapses from excitatory motoneurons to muscles and asymmetric chemical synapses from inhibitory motoneurons to muscles. The oscillations were abolished (Figs 3C and S4). These findings indicate that negative feedback is essential for oscillation generation, whereas the asymmetry of negative feedback does not modulate dorsoventral switching.

Collectively, these results imply that asymmetric inputs from premotor neurons to motoneurons, as well as asymmetric chemical synapses from motoneurons to muscles, are indispensable for coordinating dorsoventral switching. Furthermore, negative feedback is essential for the initiation of oscillations.

## The oscillations in coupled units are robustly tuned by feedback loops

Following the construction of a single functional unit, we sought to understand the coordination mechanism between two adjacent functional units in tuning oscillations. These units are generally interconnected through gap junctions and proprioception, which together form positive feedback loops. We have shown that negative feedback loops are responsible for coordinating the production of oscillations. Considering that the muscles and motoneurons of *C. elegans* exhibit high levels of coupling and synergy [24,25,43,44], we proposed that feedback loops, which include these negative and positive components, could be pivotal in modulating oscillations.

Gap junctions are distributed throughout both the head and body circuits, facilitating coordination. For instance, head muscles form gap junctions with adjacent body muscles along the worm's body, and anterior body muscles (e.g., $Muscle_{i-1}$) and motoneurons (e.g., $VB_{i-1}$) form gap junctions with posterior adjacent muscles (e.g., $Muscle_i$) and motoneurons (e.g., $VB_i$), respectively (Fig 4A). To investigate the role of these gap junctions in oscillation, we first sought solutions that enable oscillation in each functional unit using EA (S5 Fig). Subsequently, we altered the gap junction weights between adjacent units and assessed whether the adjacent muscles achieved synchronized oscillation. Our findings reveal that, in both head-body and body-body couplings, increasing the gap junction weights between muscles enhances the degree of muscle activity synchronization (Fig 4B). A similar phenomenon was observed in gap junctions between D-type motoneurons and those between SMDV-VD and SMDV-VB. Modifying the gap junction weights between these connections did not significantly affect the frequencies and amplitudes of muscle deformation (Figs 4C, 4D, and S5). These results suggest that the primary function of gap junctions, as part of the feedback loop, is to synchronize oscillations between different functional units.

In *C. elegans*, proprioception is characterized by the positive correlation between the activities of posterior motoneurons and the anterior body curvature [25], a definition that aligns with that of positive feedback in control theory. Within the body region, the ventral muscle $VM_{i-1}$ or the dorsal muscle $DM_{i-1}$ provides positive feedback to the ventral motoneuron $VB_i$ or the dorsal motoneuron $DB_i$, respectively (Fig 4E). Similarly, in the head region, the ventral or dorsal head muscles

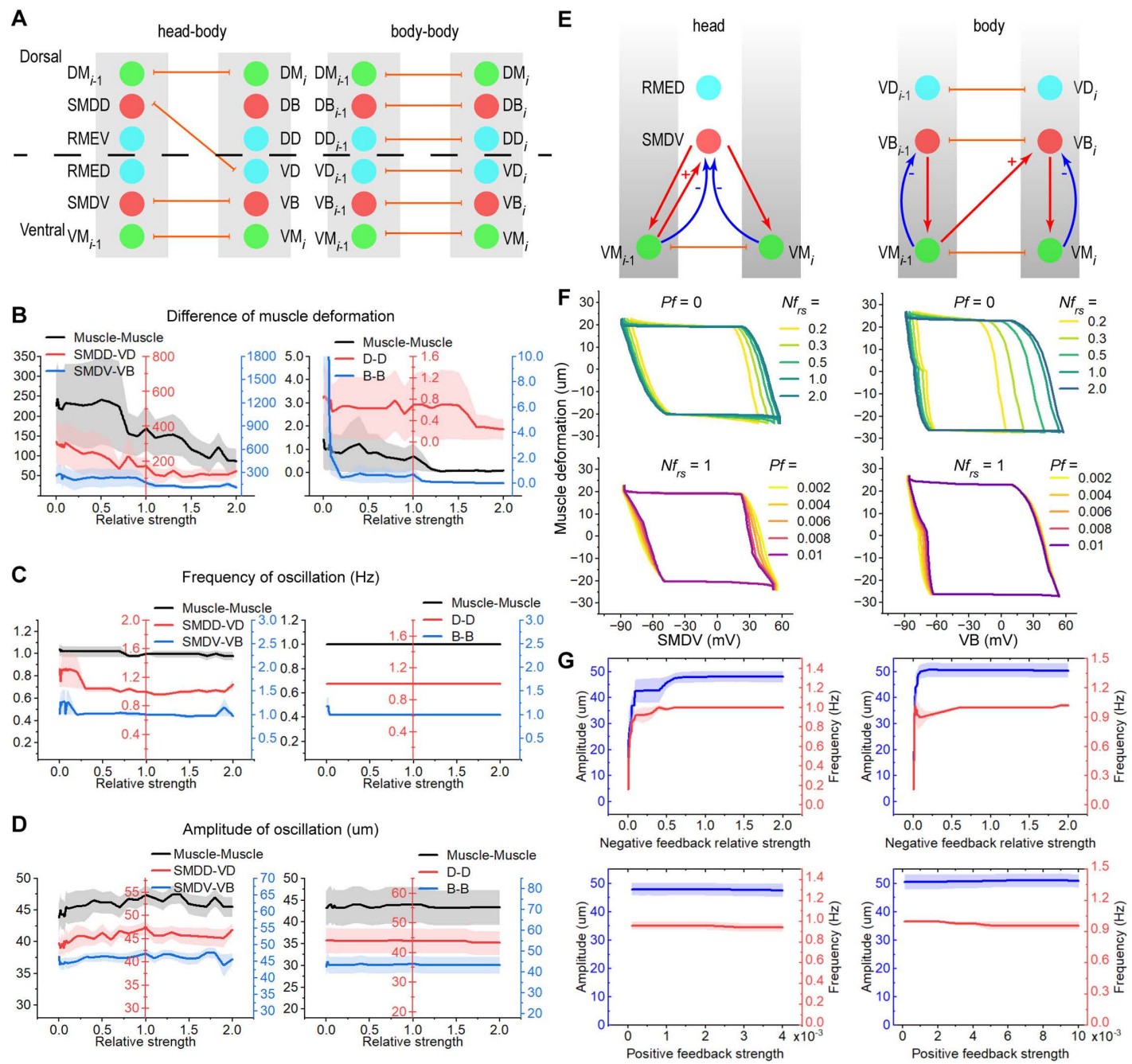

**Fig 4. Collaborative loops in the head and body of *C. elegans*.** (A) The functional units in the head and body are coupled by gap junctions between muscles, and gap junctions between motoneurons. (B) The disparity of muscle deformation between two units is influenced by gap junctions. (C) The frequencies of muscle deformation are concentrated on the target frequency. (D) The amplitudes of muscle deformation change in a small range. Error bars represent SEM (n = 5). (E) Proprioception, from muscles to motoneurons, works as positive feedback in the head and body. (F) Phase portraits by motoneurons membrane potentials and muscle deformation. $Nf_{rs}$ represents negative feedback relative strength. $Pf$ represents positive feedback strength. When the $Pf$ changes, the $Nf_{rs}$ is set as one. When the $Nf_{rs}$ changes, the $Pf$ is set as zero. (G) Frequency and amplitude when the $Nf_{rs}$ or the $Pf$ changes. Error bars are SEM (n = 5). Relative strength is calculated by $w/w_{opt}$. $w_{opt}$ is the optimized parameter.

deliver positive feedback to SMDV or SMDD, respectively. Since the functional unit incorporates negative feedback signals as well, the locomotion of *C. elegans* is regulated by a network of interconnected positive and negative feedback loops.

To investigate the modulatory role of feedback loops on oscillation, we initially employed EA to optimize parameters (S6 Fig), thereby obtaining solutions that enable the generation of oscillations. By altering the weights associated with both positive and negative feedback, we observed that varying feedback strength, from low to high, can adjust the phase trajectory of the motoneuron membrane potential and muscle deformation. Notably, positive feedback exerts an opposite effect to negative feedback which expands the trajectory from the inner circle to the outer boundary loop (Fig 4F). Increasing the strength of negative feedback enhances both the frequency and amplitude of oscillation, ultimately stabilizing at specific values. In contrast, increasing positive feedback strength maintains the frequency and amplitude of oscillation (Fig 4G). Additionally, the average percentage of stable oscillation influenced by positive feedback rises as negative feedback strength increases. (S6 Fig).

The outcomes collectively suggest that positive and negative feedback loops within functional units enhance the robustness of oscillations.

### Feedback loops generate and modulate the rhythmic behavior of simulated *C. elegans*

The findings presented above suggest that both negative and positive feedback loops are pivotal in the generation and modulation of oscillatory behaviors. To investigate whether feedback loops are responsible for generating and modulating rhythmic patterns at the whole-body level, we constructed a whole-body model by integrating 12 functional units along the worm's body. This model was developed in two distinct configurations. The first configuration employed a simplified connectome to eliminate functional redundancy between motoneurons and muscles by reducing the number of muscle cells innervated by each motoneuron. In this setup, B-type and D-type motoneurons control muscles individually, with only SMD motoneurons governing the head region (Fig 5A). The second configuration was based on the more intricate actual *C. elegans* connectome, where B-type and D-type motoneurons innervate multiple muscles (S7 Fig). In both configurations, the unknown parameters were determined using the EA optimization method (S7 Fig).

The simulation of *C. elegans* forward locomotion in a mixed Halo oil environment is depicted (S1, S2, and S3 Movies). Our findings reveal that both simulated worm types exhibit forward locomotion with the following characteristics: (i) The simulated worms display movement velocities akin to those observed in real *C. elegans*, and their rhythmic patterns align with the target frequency (S7 Fig). (ii) Heatmaps of muscle deformations demonstrate a regular progression from the head to the tail (Figs 5B, S7). (iii) The membrane potentials of the VB and DB motoneurons oscillate in an antiphase pattern (Figs 5C, S7). (iv) The gestures of the simulated worms are sinusoidal in nature, consistent with the observed movements of the real worm (Fig 5D).

To investigate the role of gap junctions, as components of feedback loops, in oscillatory dynamics, we analyzed the frequencies and amplitudes of muscle oscillations. These gap junctions facilitate communication between muscles, inhibitory motoneurons, and excitatory motoneurons. Consistent with the findings from the two functional units, our results show that enhancing gap junction weights promotes the synchronization of oscillations across the worm's body (S8 Fig). Moreover, an increase in gap junction weights is associated with not obviously changes in both the amplitude and frequency of oscillations (S8 Fig). These findings collectively suggest that gap junctions are integral to the regulation of oscillatory amplitude and frequency.

To investigate the regulatory role of feedback loops in whole-body oscillations, we adjusted the weights of both positive and negative feedback. Our study reveals that whole-body oscillations are compromised when negative feedback is absent or insufficient, highlighting the pivotal role of negative feedback in oscillation initiation. Conversely, positive feedback does not dictate the onset of oscillations but rather maintains their frequency and amplitude (Figs 5E, 5F, and S9). Furthermore, we calculated the average proportion of stable oscillations modulated by positive feedback at various levels

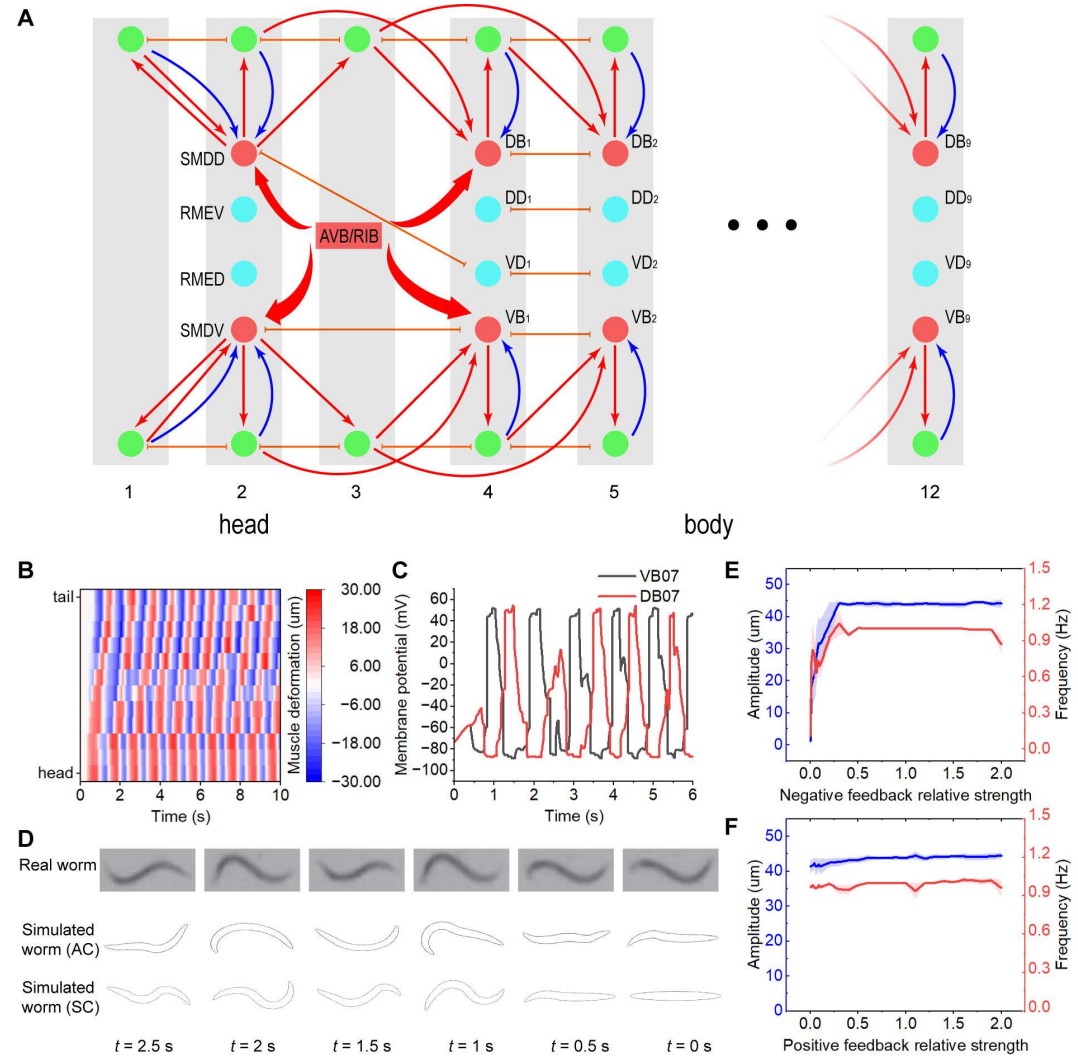

**Fig 5. Simulated *C. elegans* of the simplified connectome.** (A) Simplified connectome for *C. elegans* simulation includes the head (1st-3rd units) and body (4th-12th units). (B) Heatmap of muscle deformation. Vertical axis means body coordinate from head to tail. (C) Membrane potentials of VB7 and DB7. (D) Gestures of *C. elegans* at some specific timepoints. The upper is from the real worm. The middle is from the simulated actual-connectome worm. The lower is from the simulated simplified-connectome worm. (E) Frequency and amplitude. The horizontal axis represents the negative feedback relative strength. Error bars are SEM (n = 5). (F) Frequency and amplitude. The horizontal axis represents the positive feedback relative strength. Error bars are SEM (n = 5). Relative strength is calculated by $w/w_{opt}$. $w_{opt}$ is the optimized parameter.

of negative feedback. Our results show that the average percentage of stable oscillation increases as negative feedback strength rises (S9 Fig).

To investigate whether small parameter changes influence oscillatory behavior, we analyzed the sensitivity of all optimized variables in relation to the frequency and amplitude of oscillation. Our findings show that a one percent increase in any variable results in changes to both the frequency and amplitude of simulated *C. elegans* that are less than 15 percent. Specifically, the sensitivity of negative and positive feedback on oscillatory behavior remains below 10 percent (S9 Fig). These results demonstrate that the neuromechanical model is robust to parameter variations.

In summary, the neuromechanical model successfully simulates the movement of *C. elegans*, with negative and positive feedback loops responsible for generating and modulating the rhythmic forward locomotion of the worm.

## Discussion

*C. elegans* has emerged as a promising model organism for studying locomotion mechanisms in recent years. Both CPGs and sensory feedback loops appear to have the capability to drive the movement of the worm [16–23], with oscillators forming a continuous chain from head to tail [45]. Additionally, proprioceptive signals from muscles to motoneurons enhance our understanding of these mechanisms at the circuit level [24,25]. However, the roles of negative and positive feedback loops within this circuit remain poorly understood. Given the limitations of highly coupled spring-rod Newtonian mechanics models to address this issue, the development of a more precise neuromechanical model is essential.

In this study, we employ experimental approaches and mathematical modeling to investigate how negative and positive feedback loops within the circuit of motoneurons and muscles generate and modulate rhythmic forward locomotion. Consistent with previous studies [21–23], our data reveal that negative feedback loops effectively generate oscillations that propel *C. elegans* forward. These feedback loops, composed of motoneurons and muscles, function as distributed oscillators extending from the head to the tail. The anterior loops consist of sensory motoneurons (SMDs) and head muscles, while the middle and posterior loops are formed by B-type motoneurons and body muscles. Motoneurons activate the membrane potentials of muscles when their potential exceeds a certain threshold. In a similar manner, muscles contract only when their membrane potential surpasses this threshold. However, once muscles contract to a specified length, negative feedback currents are directed back to the motoneurons. These delays in activation generate stable oscillations. Furthermore, our data indicate some novel findings: (i) Positive feedback from muscles to motoneurons could maintain the frequency and amplitude of these oscillations. (ii) The average percentage of stable oscillation modulated by positive feedback increases with stronger negative feedback. Since the threshold for positive feedback is lower than for negative feedback, when muscles contract to this threshold, they positively activate motoneurons, thereby intensifying the contraction process and sustaining the amplitude of oscillation. Our results show that if positive feedback exceeds a certain value, oscillation ceases, which constrains the range of modulation by positive feedback. Conversely, stronger negative feedback alleviates this limitation, suggesting an important role for negative feedback in this modulation.

Here, we propose a new neuromechanical model that utilizes multibody mechanics to address the complex coupling problem of the worm body. Unlike previous Newtonian mechanics models that represent the worm body using springs and rods [18–23], our approach incorporates multiple flexible and rigid components to create a more accurate multibody mechanics system that captures the interactions between body segments. Numerical simulations demonstrate that our neuromechanical model can effectively reproduce the rhythmic forward locomotion of *C. elegans* and is suitable for analyzing feedback loops from individual functional units to the whole-body level. In this model, by integrating two negative neuron-muscle feedback loops into a single functional unit, we further elucidate that asymmetry is a prerequisite for dorsoventral switching, which is consistent with previous findings. In previous studies, the juvenile larval stage has asymmetric wiring between motoneurons and muscles [46], specifically, AS MNs generate asymmetric muscle activation, enabling bending wave propagation [47]. Our data further shows that the asymmetry of inputs from premotor neurons and the differential distribution of chemical synapses from excitatory or inhibitory motoneurons to muscles reciprocally drive dorsoventral coordinated oscillations. Moreover, the functional units of *C. elegans* from head to tail predominantly rely on gap junctions between muscles and motoneurons for coupling, which are integral components of the feedback loops. The results show that a stronger gap junction strength correlates with enhanced synchronization of oscillations. Oscillation synchronization is more effective within the normal range of gap junction strength, aligning with the observation that weak electrical couplings between motoneurons promote bending activity [39].

In summary, we present a new neuromechanical model, and our results suggest that negative feedback loops can generate sustained oscillations that drive the forward locomotion of *C. elegans*. Both negative and positive feedback loops

play roles in modulating the stability of these oscillations. Additionally, we find that the asymmetry in inputs from premotor neurons, as well as the asymmetry in the connections between motoneurons and muscles, are crucial for dorsoventral switching. The limitation of this study is that we use a graded potential model, while in reality, muscle membrane potentials generate action spikes [28,48], which could refine the simulation outcomes. And technical challenges currently limit the approach to directly verify negative feedback from muscle to motoneuron. Future experiments could use advanced techniques to address this gap. And future work could also enhance the neuromechanical model by incorporating a more detailed muscle membrane potential model and investigate whether our findings have broader implications for other biological processes involving similar competitive negative feedback loops [49–52].

## Materials and methods

### Worm strains and maintenance

All the worm strains used in this project are maintained on NGM plates seeded with *Escherichia coli* OP50 bacteria. The N2 Bristol wildtype strain and OH16230 were ordered from the Caenorhabditis Genetics Center (CGC) (Minnesota, USA). The rest transgenic worm strains were prepared via microinjection. The worms grew at 20 °C conditions during the entire experimental period. The worm strains used in this project are listed in S3 Table.

### Calcium imaging

The calcium imaging procedures were all conducted using an inverted spinning disk confocal microscope, which includes Spectral Applied Research's Diskovery spinning disk module, which runs on the Nikon Ti-E body and is controlled by Nikon NIS software.

To make the bending worms conform to the imaging ROI, mid-L4 state DNA517 worms were chosen for the imaging process. On a 9% (wt/vol, in M9 buffer) agarose gel pad, the worms were immersed in 2.5 µL of M9 buffer and then a coverslip was applied to restrain their locomotion during imaging. The whole experiment was conducted under a Nikon Plan Apochromat 20x/ 0.75 NA air objective lens with dark field illumination. Then 2 × Zyla 4.2s CMOS cameras were then used in this experiment for a 10-min dual channel (488nm and 568nm) simultaneous imaging at the scanning speed of > 1 volume/s, followed by NeuroPAL imaging for neuron identification.

To test the correlation between muscle stimulation and changes in motoneuron signals, L4 stage DNA852 worms were chosen for image processing. The red shift channelrhodopsin Chrimson was expressed on the dorsal muscle cells to be triggered by long wavelength red light. Before the experiment, the worms were anesthetized using 5mM tetramisole (CAS Number: 5086-74-8, Sigma-Aldrich). The imaging process was conducted on a CFI Apo Lambda S LWD 40XWI/ N.A. 1.15/ W.D. 0.60mm objective. The experimental procedure was designed as a "2min + 30sec + 2min" pattern, where the 2min sections refer to recording the change in GCaMP calcium signal only at 488nm, and a 30-second of 568nm light stimulation (50% intensity) was applied in the middle, and then a NeuroPAL acquisition was performed using the same parameters as previously mentioned.

### *C. elegans* forward locomotion in mixed Halo oil 27 and 700

The behaviors of *C. elegans* were recorded and analyzed using the WormLab tracking system (MBF Bioscience, VT, USA). The worm strain used in this experiment was adult N2 worms at the day 1 stage. In short, in order to create a liquid with gradient viscosity, Halocarbon oil 700 (CAS Number: 9002-83-9, Sigma-Aldrich) and Halocarbon oil 27 (CAS Number: 9002-83-10, Sigma-Aldrich) were used in the experiment, where the viscosity of the mixture liquid was represented as the percentage of the Halocarbon oil 700 (i.e., 25% Halo700 refers to a mixture of 1/4 Halocarbon oil 700 and 3/4 Halocarbon oil 27). The recording frame rate was set to 14 fps, and the total recording time for each test to 15 seconds.

To minimize the impact of OP50 on the worm body, the N2 worms were first picked onto a fresh unseeded NGM plate for temporary transition during the test. In the meantime, 50 μL of liquid was aspirated onto the center of an unseeded NGM plate with a diameter of 30 mm. One worm was picked from the transition plate and immersed into the liquid droplet. The test plate was placed under the track system, and after the worms moved forward in a 2-D direction, the recording began. After recording, the worms were removed from the droplet and another test was performed using a new droplet.

During worm locomotion, they appear in complicated typical styles: coil, spiral, twig, and swig. We focused on forward locomotion, selecting swig movies in which the worms oscillated almost on the same plane. Data was recorded online in Excel. When *C. elegans* was swimming in mixed oil, gait adaptation occurred. As the worm moved forward, the head oscillated and left a sine-wave trace. Because the growth of worms and scope scale affected the results, we normalized the worm length to 1mm. By analyzing these traces, the midpoints of the trace were selected, and the velocities and frequencies were calculated according to the time index of the midpoints.

## Divided worm according to muscle structure

In *C. elegans*, there are 95 body wall muscle cells, which are arranged as staggered pairs in four longitudinal bundles located in 4 quadrants. When the body wall of *C. elegans* is projected vertically to a two-dimensional plane, the left and right quadrants are taken as one group. Therefore, for the convenience of this study, the body wall of *C. elegans* can be divided into 12 units according to its well-patterned muscle structure. The worm could be treated as 12 connected units because of the muscle structure arranged in parallel in each bundle. The muscles corresponding to the 12 units are arranged in S1 Table. The neurons that control its forward locomotion from *C. elegans* connectome [31] were then extracted using Python. The chemical synapses and gap junctions among these motoneurons are selected to control the simulated worm. The feedback from muscles to motoneurons is set according to the connections by which motoneurons drive muscles. The negative feedback comes from the muscle that the motoneuron controls, while the positive feedback comes from the anterior two muscles that are consistent with proprioception.

## Membrane potential model

$N$ membrane potential elements, including $p$ neurons and $q$ muscles ($N = p + q$), are described as $\boldsymbol{x} = [v_{n1}, v_{n2}, \ldots, v_{np}, v_{m1}, v_{m2}, \ldots, v_{mq}]^{\mathrm{T}}$. The membrane potential models of interneurons and motoneurons are established according to Kirchhoff's current law [18,19].

$$I_{ni} + g_n(E_{leak} - v_{ni}) - C_n \dot{v}_{ni} = 0 \tag{2}$$

where $v_{ni}$ represents the membrane potential of neuron $i$ ($i = 1, 2, \ldots, p$). $g_n$, $C_n$ and $E_{leak}$ represent the leaky conductance, capacity, and equilibrium potential of neuron $i$, respectively. And the total current $I_{ni}$ is calculated as

$$I_{ni} = I_{external,ni} + I_{synapse,ni} + I_{feedback,ni} \tag{3}$$

where $I_{external,ni}$, $I_{synapse,ni}$, and $I_{feedback,ni}$ represent the input current from external, the current from synapses, and the current as feedback from muscles, to neuron $i$, respectively.

$$I_{synapse,ni} = \sum_{j=1}^{N_{gi}} w_{gap}(v_{nj} - v_{ni}) + \sum_{j=1}^{N_{ci}} w_{che} * S(v_{nj}) * (E_{Ca/K} - v_{ni}) \tag{4}$$

where $v_{nj}$ represents the membrane potential of neuron $j$, and $v_{ni}$ represents the membrane potential of neuron $i$. $w_{gap}$ represents the connected weight of the gap junction. $w_{che}$ represents connected weight of the chemical synapse. $N_{gi}$ represents the number of gap junction connected to neuron $i$. $N_{ci}$ represents the number of chemical synapses connected to

neuron $i$. $E_{Ca}$ and $E_K$ represent the equilibrium potentials of calcium and potassium ion channels, respectively. The model supposes that the synaptic activity is the sigmoidal function of the presynaptic voltage [53,54].

$$S(x) = \frac{1}{1 + e^{-\frac{x-V_x}{\theta_x}}}$$

(5)

where $V_x$ represents the bias value that shifts the output range (namely threshold of activation) and $\theta_x$ adjust the sensitivity of output (namely time constant of activation).

Similarly, according to Kirchhoff's current law, an action potential model of muscle could be set up.

$$I_{synapse,mj} + g_m(E_{leak} - v_{mj}) - C_m\dot{v}_{mj} = 0$$

(6)

Where $v_{mj}$ represents the membrane potential of muscle $j$ ($j$ = 1, 2, ..., $q$). $C_m$ and $g_m$ represent the capacity and leak conductance of muscle membrane, respectively. $E_{leak}$ represents the leak potential, the same as neurons. $I_{synapse,mi}$ is defined similarly with $I_{synapse,ni}$.

## Kinematic recursive relationships

We assumed that the entire worm body was divided into 13 T-shaped segments, which were connected by 12 pairs of muscles. The geometric relationship between adjacent segments is shown in S10 Fig. The segment $j$ is anterior to segment $i$, where $i$ = 1, 2, ..., 13, and $j$ = $i$ - 1. $C_\alpha$ ($\alpha$ = i, j) is the center of mass of the segment. The adjacent segments are connected by the revolute joint, and points $P$ and $Q$ represent the coincident connection points on the two segments.

The generalized velocity of segment $i$ is defined as

$$v_i = \begin{bmatrix} \dot{r}_i^T & \omega_i \end{bmatrix}^T$$

(7)

where $\mathbf{r}_i$ represents the position vector of the center of mass of segment $i$, the dot symbol represents the derivative with respect to time, and $\omega_i$ is the angular velocity of segment $i$.

The Newton-Euler equations of locomotion can be expressed as

$$\begin{bmatrix} m_i I_2 & \\ & J_i \end{bmatrix} \begin{bmatrix} \ddot{r}_i \\ \dot{\omega}_i \end{bmatrix} = \begin{bmatrix} F_i \\ T_i \end{bmatrix}$$

(8)

where $m_i$ and $J_i$ represent the mass and moment of inertia of segment $i$, respectively. $\mathbf{I_2}$ represents a 2x2 identity matrix. $\mathbf{F}_i$ and $T_i$ represent the external force and torque of segment $i$, respectively.

The angular velocity and acceleration relationships between adjacent segments are linear and can be obtained as follows

$$\begin{cases} \omega_i = \omega_j + \dot{\theta}_i \\ \dot{\omega}_i = \dot{\omega}_j + \ddot{\theta}_i \end{cases}$$

(9)

The position vector of segment $i$ can be expressed as

$$r_i = r_j + \rho_j^Q - \rho_i^P$$

(10)

By taking the derivatives of Eq 10, the translational velocity and acceleration relationships between adjacent segments are obtained

$$\begin{cases} \dot{r}_i = \dot{r}_j + \tilde{I}\left(\rho_j^Q - \rho_i^P\right)\omega_j - \tilde{I}\rho_i^P\dot{\theta}_i \\ \ddot{r}_i = \ddot{r}_j + \tilde{I}\left(\rho_j^Q - \rho_i^P\right)\dot{\omega}_j - \tilde{I}\rho_i^P\ddot{\theta}_i - \rho_j^Q\omega_j^2 + \rho_i^P\omega_i^2 \end{cases} \tag{11}$$

where an auxiliary matrix is defined as

$$\tilde{I} = \begin{bmatrix} 0 & -1 \\ 1 & 0 \end{bmatrix} \tag{12}$$

Now, the recursive relationships of velocity and acceleration between adjacent segments can be expressed as

$$\begin{cases} v_i = T_{ij}v_j + U_i\dot{\theta}_i \\ \dot{v}_i = T_{ij}\dot{v}_j + U_i\ddot{\theta}_i + \beta_i \end{cases} \quad (i = 2, ..., M) \tag{13}$$

where

$$T_{ij} = \begin{bmatrix} I & \tilde{I}\left(\rho_j^Q - \rho_i^P\right) \\ 0 & 1 \end{bmatrix}, \quad U_i = \begin{bmatrix} -\tilde{I}\rho_i^P \\ 1 \end{bmatrix}, \quad \beta_i = \begin{bmatrix} -\rho_j^Q\omega_j^2 + \rho_i^P\omega_i^2 \\ 0 \end{bmatrix} \tag{14}$$

The definitions for the first segment are provided as $U_1 = I_3$, $\beta_1 = 0$. According to Eq 13, the relationships between the generalized velocity of segment $i$ and the time derivatives of generalized coordinates are

$$\begin{cases} v = G\dot{y} \\ \dot{v} = G\ddot{y} + \hat{g} \end{cases} \tag{15}$$

Where $v = [v_1^T, v_2^T, ..., v_{13}^T]^T$, $y = [r_1^T, \theta_1^T, ..., \theta_{13}^T]^T$ and

$$G_{ik} = \begin{cases} T_{ij}G_{jk} & \text{if } k < i \\ U_i & \text{if } k = i \\ 0 & \text{if } k > i \end{cases} \quad (i, k = 1, ..., M), \quad g_{ik} = \begin{cases} T_{ij}g_{jk} & \text{if } k < i \\ \beta_i & \text{if } k = i \\ 0 & \text{if } k > i \end{cases} \quad (i, k = 1, ..., M) \tag{16}$$

$$G = \begin{bmatrix} G_{11} & \cdots & G_{1M} \\ \vdots & & \vdots \\ G_{M1} & \cdots & G_{MM} \end{bmatrix}, \quad g = \begin{bmatrix} g_{11} & \cdots & g_{1M} \\ \vdots & & \vdots \\ g_{M1} & \cdots & g_{MM} \end{bmatrix} \tag{17}$$

The summation of $g$ is expressed as

$$\hat{g} = \begin{bmatrix} \hat{g}_1^T & \cdots & \hat{g}_{13}^T \end{bmatrix}^T, \quad \hat{g}_i = \sum_k g_{ik} \tag{18}$$

### Driven force and feedback

When the membrane potential is higher than the threshold, the muscle contracts. We set up the sigmoid function to describe this process, which is formed in the same way as Eq 5.

$$f_{muscle,mi} = w_m S(v_{mi})(L_{min} - l_i) \tag{19}$$

where $v_{mi}$ represents the membrane potential of muscle $i$, and $l_i$ represents the length of muscle $i$. $L_{min}$ represents the shortest length of one muscle. $w_m$ represents the factor of driving effect from membrane potential to the length of the muscle. This model also supposes that the driving force is the sigmoidal function of presynaptic voltage, but $V_x$ and $\theta_x$ are different from chemical synapses in Eq 5.

$$I_{feedback,ni} = \begin{cases} w_f S(\delta_j)(E_{Ca/K} - v_{ni}), & \delta_j > 0 \\ w_f(1 - S(\delta_j))(E_{Ca/K} - v_{ni}), & \delta_j < 0 \end{cases} \tag{20}$$

$$\delta_j = L - l_j \tag{21}$$

where $v_{ni}$ represents the membrane potential of the neuron $i$. $\delta_j$ represents deformation of muscle $j$, $L$ represents the resting length of one muscle, and $l_j$ represents the length of muscle $j$. $w_f$ represents the weight of the feedback effect. $E_{Ca}$ nd $E_K$ have the same definition as the above. When the muscle deformation is greater than 0, the positive feedback is set to open the calcium ion channel. When the muscle deformation is smaller than 0, the positive feedback is set to open potassium channels. On the contrary, the negative feedback is set to open the potassium ion channel when the muscle deformation is greater than 0 or the calcium ion channel when the muscle deformation is smaller than 0. Although positive feedback and negative feedback are similarly defined in form, they have different thresholds $V_x$, which are optimized by the EA.

## Passive force of muscle

The expressions of the elastic and damping force are derived as follows. The dorsal is represented by "$D$" and the ventral is represented by "$V$" in the subscript. As shown in S10 Fig, the vector of the spring can be expressed as

$$s_{mi} = \rho_i^Q + b_{m(i+1)} - \rho_i^P - b_{mi} \quad (m = D, V) \tag{22}$$

Taking the derivative of Eq 22, the velocity of the spring vector can be expressed as

$$\dot{s}_{mi} = \tilde{I}\left(\rho_i^Q \omega_i + b_{m(i+1)}\omega_{i+1} - \rho_i^P \omega_i - b_{mi}\omega_i\right) \tag{23}$$

The elastic and damping force can be calculated by

$$f_{mi}^k = k\left(\left|s_{mi}\right| - L\right), \quad f_{mi}^c = c\dot{s}_{mi}^{\mathrm{T}}\frac{s_{mi}}{\left|s_{mi}\right|} \tag{24}$$

where $L$ represents the origin length of the spring. $k$ and $c$ represent the spring and damping constants of the muscle, respectively.

## Environment properties

In low Reynolds numbers liquid, the viscous force from the environment has the linear relationship with the velocity in the tangent and normal directions. Then environmental force at the centroid of segment $i$ is as follows.

$$\begin{cases} f_{\perp,i} = C_n v_{\perp,i} \\ f_{\parallel,i} = C_t v_{\parallel,i} \end{cases} \tag{25}$$

where $f_{\perp,i}$ and $f_{\|,i}$ represent the normal and tangent direction force to segment $i$, respectively. $C_n$ and $C_t$ represent the normal and tangent viscous coefficients, respectively. $v_{\perp,i}$ represents the velocity of the normal component, and $v_{\|,i}$ represents the velocity of the tangent component.

**Multibody dynamic model**

The dynamic equations of one rigid segment $i$ are the Newton-Euler equations.

$$-M_i \dot{v}_i + f_i = 0 \tag{26}$$

where $M_i$ represents the mass matrix of segment $i$. $v_i$ represents the generalized velocity of segment $i$. $f_i$ represents the external force of segment $i$. The external force consists of the active and passive muscle forces, which can be given as follows:

$$f_i = \begin{bmatrix} \left(f_{muscle,Di} + f^K_{Di} + f^C_{Di}\right) \frac{s_{Di}}{|s_{Di}|} + \left(f_{muscle,Vi} + f^K_{Vi} + f^C_{Vi}\right) \frac{s_{Vi}}{|s_{Vi}|} - f_{\perp,i} - f_{\|,i} \\ \rho^T_{Di} \tilde{I} \left(f_{muscle,Di} + f^K_{Di} + f^C_{Di}\right) \frac{s_{Di}}{|s_{Di}|} + \rho^T_{Vi} \tilde{I} \left(f_{muscle,Vi} + f^K_{Vi} + f^C_{Vi}\right) \frac{s_{Vi}}{|s_{Vi}|} \end{bmatrix} \tag{27}$$

$$\rho_{mi} = \rho^P_i + b_{mi} \, (m = D, V) \tag{28}$$

Based on Jourdain's variation principle [55], the dynamic equations of the multibody system can be written as

$$\sum_i \Delta v_i^T \left(-M_i \dot{v}_i + f_i\right) = 0 \tag{29}$$

The variation of the generalized velocity based on Eq 15 can be expressed as

$$\Delta v_i = G_i \Delta \dot{y}_i \tag{30}$$

By substituting the relationships between the generalized velocity of segment $i$ and the derivatives of the generalized coordinate Eq 15 into Eq 29, the dynamic equations of the multibody system can be rewritten as

$$Z\ddot{y} = z \tag{31}$$

where the generalized mass matrix and generalized force are

$$Z = \sum_i G_i^T M_i G_i, \quad z = \sum_i G_i^T \left(f_i - M_i \hat{g}_i\right) \tag{32}$$

**Evolutionary algorithm**

The parameters required for all connection modes in the neuromechanical model are obtained by using genetic algorithm in MATLAB. The algorithm begins with randomly generated vectors that encode unknown parameters. Every individual in the population is assigned the fitness to evaluate their locomotion performance. After a series of genetic operations such as selection, crossover, and mutation, a new generation is produced. As the entire process is repeated, the population would converge on well-fitted individuals. The optimized parameters include connection weights of chemical synapses and gap junctions, time constants of neurons and muscles, the weights of positive and negative feedback, thresholds for two types of feedback, spring and damping constants of muscle, and viscosity factors of the environment.

The fitness function with two components is established to evaluate whether the parameters of the neuromechanical model could generate the rhythmic pattern for forward locomotion consistent with the oscillation frequency and locomotion speed in the mixed oil. The first component is to ensure rhythmic activities that match the frequency observed in forward *C. elegans* [18, 19, 20].

$$F_1 = \sum_{i=1,2,\ldots,q} \left( \frac{A*T}{2\int_0^T \left|\frac{dS_i}{dt}\right|dt} + \left|1 - \frac{fre_i}{fre_a}\right| \right)$$

(33)

Where $A$ represents the amplitude threshold ($A = 0.5$), $S_i$ represents the deformation of muscle $i$, $T$ represents the simulation time, $fre_i$ represents the frequency of deformation muscle $i$ and $fre_a$ represents the observed oscillation frequency, which is set as 1 Hz. Since the GA toolbox in MATLAB looks for the minimum fitness function, the fitness function was written in the form of the cumulative sum.

The second component encourages the simulated worm to move forward matching velocity in mixed oil [18, 19, 20].

$$F_2 = \left|1 - \frac{V}{V_a}\right|$$

(34)

where $V$ represents the average forward velocity of the simulated worm, and $V_a$ represents the average forward movement velocity (126 um/s) in the oil mixture. The whole fitness function is the sum of these two components for simulating whole *C. elegans*.

## Difference of muscle deformation

To evaluate the synchronization of muscles, we calculated the difference of muscle deformation by following formation.

$$D_m = \left( \sum_{i,j=1,2,\ldots,M} \overline{\sum (\delta_i - \delta_j)^2} \right) / C_M^2, i < j$$

(35)

Where $D_m$ represents the difference of deformation of $M$ muscles. $\delta_i$ and $\delta_j$ represent the muscle deformation time series of muscles $i$ and $j$, respectively. The difference is calculated by any two muscles' deformation. Since muscle deformation presented the stable rhythmic pattern, we used the average value to evaluate the difference.

## Sensitivity analysis

To evaluate the robustness of the neuromechanical model, we employed sensitivity analysis from all variables to frequency and amplitude by following formation.

$$S(f, x_i) = \frac{\Delta f/f}{\Delta x_i/x_i}$$

(36)

Where $S(f, x_i)$ is the sensitivity from $x_i$ to $f$. $f$ is selected from frequency and amplitude. All the values of sensitivity represent whether frequency or amplitude is robust to optimized variable perturbation. Specifically, we checked the sensitivity of two types of feedback to frequency and amplitude.

## Supporting information

**S1 Fig. Locomotion properties of *C. elegans* in mixed Halo oil.** (**A**) Frequency of *C. elegans* rhythmic forward locomotion. Each box represents mean and SEM from n ≥ 5 worms. (**B**) Velocity of *C. elegans* rhythmic forward locomotion. Error bars are SEM.
(TIF)

**S2 Fig. Calcium activities of muscles of bending worm and cross-correlations between muscle and motoneuron.** (**A**) Calcium activities of dorsal and ventral muscles. (**B**) Cross-correlations between dorsal and ventral muscles. (**C**) Cross correlations between dorsal muscle and motoneuron VB7. (**D**) Calcium activities of dorsal and ventral muscles. (**E**) Cross correlations between dorsal and ventral muscles. (**F**) Cross correlations between dorsal muscle and motoneuron VB7. (**G**) Cross correlations between dorsal muscle and motoneuron DB5. (**H**) Cross correlations between ventral muscle and motoneuron VB7. (**I**) Cross correlations between ventral muscle and motoneuron DB5. (**J**) Cross correlations between dorsal muscle and motoneuron DB5. (**K**) Cross correlations between ventral muscle and motoneuron VB7. (**L**) Cross correlations between ventral muscle and motoneuron DB5. Worms bend to the dorsal in **A, B, C, G, H** and **I**. Worms bend to the ventral in **D, E, F, J, K,** and **L**. Error bars are SEM (n = 10).
(TIF)

**S3 Fig. Frequency and amplitude modulation analysis.** Negative feedback, weight of motoneuron to muscle, and muscle contraction ability modulate oscillation frequency and amplitude. Error bars are SEM (n = 5). Relative strength is calculated by $w/w_{opt}$. $w_{opt}$ is the optimized parameter.
(TIF)

**S4 Fig. Optimized results and heatmaps of muscle deformation.** (**A**) Optimization process by EA. (**B**) Heatmap of muscle deformation. The vertical axis represents the relative strength of the chemical synapse from EMN to IMN between dorsal and ventral sides. (**C**) Heatmap of muscle deformation. The vertical axis represents the relative strength of the chemical synapse from IMN to IMN between dorsal and ventral sides. (**D**) Heatmap of muscle deformation. The vertical axis represents the relative strength of the negative feedback between dorsal and ventral sides. (**E**) Heatmap of muscle deformation. The vertical axis represents the relative strength of the chemical synapse from IMN to muscle between dorsal and ventral sides. Three types of connections, synapses from excitatory to inhibitory motoneurons, synapses between inhibitory motoneurons, and negative feedback are mock ablated, respectively. Relative strength is calculated by $w_d/w_v$. $w_v$ is the optimized parameter at ventral side.
(TIF)

**S5 Fig. Optimization process and heatmap of muscle deformation.** (**A**) Optimization process by EA. The left is for the head-body part, right is for the body-body part. (**B**) Heatmaps of muscle deformation by changing the strength of three types of gap junctions in the head-body coupling. (**C**) Heatmaps of muscle deformation by changing the strength of three types of gap junctions in body-body coupling. Relative strength is calculated by $w/w_{opt}$. $w_{opt}$ is the optimized parameter.
(TIF)

**S6 Fig. Optimization processes and average percentage of stable oscillation.** (**A**) Optimization by EA. The left is for the head-part, the right is for the body-part. (**B**) Average percentage of stable oscillation modulated by positive feedback. The horizontal axis represents five levels of negative feedback relative strength.
(TIF)

**S7 Fig. Simulated actual-connectome *C. elegans*.** (**A**) Connections from actual connectome are used to simulate *C. elegans*. (**B**) Optimization process by EA. The left is for the simplified connectome, the right is for the actual connectome. (**C**) Velocity of *C. elegans* locomotion. Error bars are SEM (n = 6, 10,10, respectively). (**D**) Frequency of *C. elegans* locomotion. Error bars are SEM (n = 6, 10,10, respectively). (**E**) Heatmap of muscles deformation. (**F**) Membrane potentials of VB7 and DB5.
(TIF)

**S8 Fig. Electrical synapse analysis for two types of simulated *C. elegans*.** (**A**) Average difference among muscle deformation. (**B**) Frequencies of oscillation. (**C**) Amplitudes of oscillation. (**D**) Average difference among

muscle deformation. (**E**) Frequencies of oscillation. (**F**) Amplitudes of oscillation. **A**, **B**, and **C** are from simulated simplified-connectome *C. elegans*. **D**, **E**, and **F** are from simulated actual-connectome *C. elegans*. Relative strength is calculated by $w/w_{opt}$. $w_{opt}$ is the optimized parameter.
(TIF)

**S9 Fig. Oscillation modulated by feedback and sensitivity analysis for frequency and amplitude.** (**A**) Frequency and amplitude. The horizontal axis represents the negative feedback relative strength. Error bars are SEM (n=5). (**B**) Frequency and amplitude. The horizontal axis represents the positive feedback relative strength. Error bars are SEM (n=5). **A** and **B** are from simulated actual-connectome *C. elegans*. (**C**) Average percentage of stable oscillation modulated by positive feedback. The horizontal axis represents five levels of negative feedback relative strength. (**D**) Sensitivity analysis from all variables to frequency and amplitude. Negative feedback is marked as blue column. Positive feedback is marked as red column. Other variables are marked as grey column. The left is for simplified-connectome *C. elegans*. The right is for actual-connectome *C. elegans*. Relative strength is calculated by $w/w_{opt}$. $w_{opt}$ is the optimized parameter.
(TIF)

**S10 Fig. Mechanics relationship.** (**A**) Physical notion. (**B**) Geometry relationship between adjacent segments. (**C**) Vectors of springs in two T-shaped rigid rods.
(TIF)

**S1 Table. Divided worm into 12 parts in the two-dimensional plane.** Muscle structure could be divided into 12 segments when treated as two-dimensional. Since two muscles are arranged in parallel in one unit, such as vBWM01, and vBWM02, one unit is set as one dorsal muscle and one ventral muscle. The data was handled using Python.
(DOCX)

**S2 Table. Parameters of neurons and muscles.** The whole worm length is normalized to 1000 um, then the rest length of muscle is that whole worm length divides 13 (represents the number of rigid rods). Other parameters are set from references. Reference capacity and conductance are not used directly but rather multiply factors, which are optimized by EA.
(DOCX)

**S3 Table. Worm strains. Four types of worm strains were used in experimental studies.** We observed and recorded worm behaviors in mixed oil by N2. OH16230 were ordered from the Caenorhabditis Genetics Center (CGC) (Minnesota, USA). DNA517 worms were chosen for the imaging process to fit the bending worms in the imaging ROI. To test whether muscle stimulation influenced motoneurons, DNA852 worms were chosen.
(DOCX)

**S1 Movie. *C. elegans* moved forward in 50% Halo 700 mixed oil** Properties of *C. elegans* moved forward in 50% Halo 700 mixed oil were used for optimization. We recorded the forward locomotion of the worm in this group of mixed oil for comparison.
(MP4)

**S2 Movie. Simulated *C. elegans* from the simplified connectome moved forward.** We simulated the worm from the simplified connectome with 42 neurons, and 24 muscles by MATLAB to fit the target rhythm 1 Hz and speed (126 um/s).
(MP4)

**S3 Movie. Simulated *C. elegans* from the actual connectome moved forward.** We simulated the worm from the actual connectome for forward locomotion with 45 neurons, and 24 muscles by MATLAB to fit the target rhythm 1 Hz and speed (126 um/s).
(MP4)

## Acknowledgments

We thank Yugen Xi, who is a professor at the Department of Automation at Shanghai Jiao Tong University, for his help and suggestions. The computations in this work were carried out on the cloud host supported by the Center for High-Performance Computing at Shanghai Jiao Tong University.

## Author contributions

**Conceptualization:** PENG ZHAO, Boyang Wang, Yi Rong, Ye Yuan, Jian Liu, Hong Huo, Zhuyong Liu, Zhaoyu Li, Tao Fang.

**Formal analysis:** PENG ZHAO, Boyang Wang, Yi Rong.

**Funding acquisition:** Zhuyong Liu, Zhaoyu Li, Tao Fang.

**Methodology:** PENG ZHAO, Boyang Wang, Yi Rong, Ye Yuan, Jian Liu, Zhuyong Liu, Zhaoyu Li, Tao Fang.

**Project administration:** PENG ZHAO.

**Software:** PENG ZHAO, Boyang Wang.

**Supervision:** Tao Fang.

**Validation:** PENG ZHAO, Yi Rong.

**Visualization:** PENG ZHAO, Boyang Wang.

**Writing – original draft:** PENG ZHAO, Boyang Wang, Yi Rong.

**Writing – review & editing:** PENG ZHAO, Boyang Wang, Yi Rong, Ye Yuan, Jian Liu, Hong Huo, Zhuyong Liu, Zhaoyu Li, Tao Fang.

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
