## [Decision Letter · Decision Letter 0]

PCOMPBIOL-D-24-02033

The roles of feedback loops in the Caenorhabditis elegans rhythmic forward locomotion

PLOS Computational Biology

Dear Dr. Fang,

Thank you for submitting your manuscript to PLOS Computational Biology. After careful consideration, we feel that it has merit but does not fully meet PLOS Computational Biology's publication criteria as it currently stands. Therefore, we invite you to submit a revised version of the manuscript that addresses the points raised during the review process.

Please submit your revised manuscript within 60 days Apr 13 2025 11:59PM. If you will need more time than this to complete your revisions, please reply to this message or contact the journal office at ploscompbiol@plos.org. Please include the following items when submitting your revised manuscript:

We look forward to receiving your revised manuscript.

Kind regards,

Adriana San Miguel

Academic Editor

PLOS Computational Biology

Marc Birtwistle

Section Editor

PLOS Computational Biology

**Additional Editor Comments :**

Please address the points raised by all reviewers and clarify the unique contribution of the work in the context of prior studies. Particularly, address the significance of the model ensuring the consensus understanding is supported by current literature, as described by reviewer 2.

**Journal Requirements:**

- ® on page: 33.

3) Please note that the section "Methods" should be included in the main file of the manuscript. Please ensure all required sections are present and in the correct order. Make sure section heading levels are clearly indicated in the manuscript text, and limit sub-sections to 3 heading levels. An outline of the required sections can be consulted in our submission guidelines here:

5) We notice that your supplementary Figures, Tables, and information are included in the manuscript file. Please remove them and upload them with the file type 'Supporting Information'. Please ensure that each Supporting Information file has a legend listed in the manuscript after the references list.

6) In the online submission form, you indicated that "All data and code supporting the findings of this study are available from the corresponding author (Tao Fang) upon request."  All PLOS journals now require all data underlying the findings described in their manuscript to be freely available to other researchers, either

1. In a public repository

2. Within the manuscript itself

3. Uploaded as supplementary information.

7) Thank you for stating that "All data and code for this article have been deposited on Zenodo at "https://doi.org/10.5281/zenodo.12746051." This link reaches a DOI Not Found page. Please amend this to a working link or provide further details to locate the data.

8) Please amend your detailed Financial Disclosure statement. This is published with the article. It must therefore be completed in full sentences and contain the exact wording you wish to be published.

9) Please ensure that the funders and grant numbers match between the Financial Disclosure field and the Funding Information tab in your submission form. Note that the funders must be provided in the same order in both places as well. Currently, "Li lab startup funding" is missing from the Funding Information tab.

Please indicate by return email the full and correct funding information for your study and confirm the order in which funding contributions should appear. Please be sure to indicate whether the funders played any role in the study design, data collection and analysis, decision to publish, or preparation of the manuscript.

**Reviewers' comments:**

Reviewer's Responses to Questions

Reviewer #1: The manuscript by Zhao et al. investigates how C. elegans generates and tunes rhythmic forward locomotion. By combining calcium imaging, theoretical neuromechanical modeling, and connectome-based simulations, the authors argue that both positive and negative feedback loops can flexibly tune oscillatory patterns. In contrast, they find that a canonical central pattern generator alone cannot fully account for the worm’s flexibility in rhythmic movement. Instead, they propose that negative neuron-muscle feedback loops—augmented by positive feedback—govern oscillation amplitude and frequency, thereby enabling a broader range of behaviors. This study has several strengths: 1. The paper clearly demonstrates how negative feedback can generate oscillations, while positive feedback and gap junctions modulate frequency and amplitude. The distinction between dorsal and ventral loops, and the requirement for asymmetry, adds mechanistic depth to the model. 2. It integrates calcium imaging in both immobilized and freely moving worms with a neuromechanical model. Linking experimental data to circuit-based simulations allows for a detailed analysis of how muscle and neuron feedback might drive rhythmic body bending. 3. By emphasizing feedback loops that incorporate muscle deformation and proprioceptive signals, this work provides a valuable perspective on the flexibility of rhythmic movements, which will interest neuroscientists studying motor control in simpler model systems. Overall, this is an excellent manuscript, and I support its publication.

One concern, however, relates to the discussion of central pattern generators (CPGs). While the authors present strong evidence favoring a feedback-based mechanism between motoneurons and muscles, they do not fully rule out the possibility that upstream CPGs could exist in C. elegans. Conclusively eliminating a CPG would require more direct in vivo manipulations, which is not the focus of this study in my opinion. Accordingly, I recommend that the authors revise lines 182–184, 384–386, and 394–396 or teo expand discussions to acknowledge how CPGs might still contribute. This is a minor concern, but this revision would clarify that while their model strongly supports muscle-driven oscillations, partial or upstream CPG mechanisms cannot yet be entirely excluded.

Reviewer #2: The article is well written and well presented. The authors develop a neuromechanical model of C. elegans which includes an idealized body and neural circuit constrained by known neuroanatomy.

What is not clear with this article is the significance of the insights gained from the model, and how it is set apart from previous work. The manuscripts cites a lot of the relevant previous work, but it does not concretely discuss the way in which the insights gained by their model agree with or contradict any of the existing models. This is particularly puzzling because it would seem as if at least a number of their claims were being supported by previous models, including prominently the important role of proprioceptive feedback in locomotion (in contrast to hypothesis suggesting central pattern generators). Counterintuitively, the authors make the narrative so as to seem as if the prevailing view of C. elegans locomotion is that it is controlled by CPGs, and they claim to argue that their model suggests feedback loops instead. However, any reasonable survey of the literature on C. elegans forward locomotion reveals that the large majority of computational models, perspectives, and experimental works favor the feedback hypothesis, from as far back as it has been studied.

Reviewer #3: Summary

The manuscript presents a novel re-evaluation of C. elegans locomotion, arguing that rhythmic movement arises from feedback loops rather than a purely central pattern generator (CPG)-driven mechanism. The authors employ calcium imaging, optogenetics, and computational neuromechanical modeling to show that: 1) Negative feedback loops between motoneurons and muscles play a key role in generating oscillations; 2) Gap junctions synchronize activity, and proprioception fine-tunes oscillatory behavior; and 3) A computational model integrates known neuronal connections, biomechanical forces, and proprioceptive feedback to simulate movement dynamics. These findings challenge the longstanding CPG-centric view and propose that locomotion rhythms emerge from neuromuscular interactions rather than intrinsic neuronal oscillations alone. The study is timely and innovative and addresses fundamental questions in neuromechanical control.

Some key claims require additional validation, and the modeling approach needs further robustness testing. While the study presents strong experimental and computational evidence, it overstates the role of feedback loops at the expense of CPGs and lacks sufficient biological validation for key claims. The model requires sensitivity analysis to assess robustness, and additional experimental validation is needed to confirm that negative feedback loops actively drive oscillations.

Specific issues

1. Overstated rejection of CPGs. The study presents feedback loops as an alternative to CPGs, but prior work suggests they work together rather than being mutually exclusive. Previous research demonstrated that neuromodulation (e.g. serotonin and dopamine) regulates gait transitions, implying that central circuits still contribute to rhythmicity. More recent work by Du et al. (2025) showed that body-wall muscle excitability is influenced by intrinsic biophysical properties, suggesting muscle feedback may play a role, but not in isolation. It seems prudent to acknowledge that CPGs and feedback loops likely co-exist, rather than presenting feedback loops as the sole oscillation mechanism. Additionally, the authors should discuss how neuromodulation and proprioception interact to regulate rhythmicity and really do a more thorough job discussing/evaluating/incorporating the wealth of work conducted in this field (particularly previous efforts to point out the importance of feedback loops).

2. Robustness. The neuromechanical model relies on parameter optimization (via an evolutionary algorithm), but no sensitivity analysis is provided. It is unclear whether small parameter changes significantly affect oscillatory behavior. The authors could for example perform sensitivity analysis to demonstrate robustness to parameter perturbations (e.g., synaptic weights, proprioceptive delay).

3. Lack of direct experimental support for negative feedback loops. While calcium imaging shows that B-type motoneurons require proprioceptive input, this does not directly confirm that negative feedback loops are responsible for oscillation generation. There are no electrophysiological recordings from motoneurons showing feedback inhibition. This is understandable given the experimental challenged posed by said recordings. Still, the authors could have (for example) conducted laser ablation of muscles to determine if removing proprioceptive feedback disrupts oscillations. Alternatively (in in addition) they could have tested genetic mutants that are defective at proprioception (e.g., mec-4, unc-54) to determine whether rhythmicity is affected.

4. Integration with current electrophysiological models of C. elegans muscles. This comment is not entirely fair since it refers to a work that just came out. But in future version the authors could look at a new and apparently pertinent work by Du et al. (2025). That study provides a biophysically detailed model of C. elegans muscle membrane dynamics, showing that muscles exhibit burst-firing properties with intrinsic frequency tuning. In light of this new work, the present study could incorporate these findings to further support their argument that muscle feedback contributes to oscillatory patterns. The manuscript might benefit from comparing muscle bursting properties observed in Du et al. (2025) to the oscillatory patterns in their current model. They could then discuss whether muscle excitability contributes to rhythmic behavior independently of motoneuron input.

5. Functional interpretation of oscillatory zones. The manuscript defines three oscillatory zones, but does not relate them to real-world C. elegans behavior. Previous work showed that locomotion transitions between swimming and crawling involve neuromodulatory regulation, implying that these oscillatory modes may correspond to behavioral states. More recently, work by Fazyl et al. (2025) suggests that muscle activation differs fundamentally between swimming and crawling animals. This offers an opportunity for the authors to test their assumptions and model. At minimum I would suggest they provided experimental data showing how these three oscillatory zones correlate with actual movement patterns (e.g., response to viscosity changes, transitions between crawling and swimming).

**Have the authors made all data and (if applicable) computational code underlying the findings in their manuscript fully available?**

Reviewer #1: Yes

Reviewer #2: **No: ** Data and code are not available to the public; the authors state that data and code is available upon request.

Reviewer #3: **No: ** The link to data and code provided (https://doi.org/10.5281/zenodo.12746051) does not work. This was an issue.

PLOS authors have the option to publish the peer review history of their article (what does this mean? ). If published, this will include your full peer review and any attached files.

**Do you want your identity to be public for this peer review?** For information about this choice, including consent withdrawal, please see our Privacy Policy .

Reviewer #1: No

Reviewer #2: No

Reviewer #3: No

**Figure resubmission:**
---

## [Decision Letter · Decision Letter 1]

Dear Prof. Fang,

We are pleased to inform you that your manuscript 'The roles of feedback loops in the Caenorhabditis elegans rhythmic forward locomotion' has been provisionally accepted for publication in PLOS Computational Biology.

Best regards,

Adriana San Miguel

Academic Editor

PLOS Computational Biology

Marc Birtwistle

Section Editor

PLOS Computational Biology

Reviewer's Responses to Questions

**Comments to the Authors:**

Reviewer #1: I appreciate the revisions the authors have made, particularly the more balanced presentation of central pattern generators (CPGs) and feedback loop mechanisms. All of my concerns have been fully addressed in the revised manuscript.

Reviewer #3: The authors made significant revisions that improved the manuscript’s clarity, balance, and scientific rigor. They softened earlier claims about feedback loops being the sole source of oscillations and now acknowledge that central pattern generators (CPGs) may also contribute. This more balanced view is reflected throughout the revised manuscript.

To address concerns about novelty, the authors expanded comparisons with earlier models and highlighted key innovations, such as incorporating multibody mechanics and modeling both negative and positive feedback loops. They positioned their work as an extension of existing feedback theories rather than a rejection of CPGs.

They also added sensitivity analyses to show the model’s robustness and integrated recent muscle dynamics research to support their results. While they admitted the lack of direct experimental validation—particularly for negative feedback loops—they outlined future experiments to address this. They clarified how their modeled oscillatory zones relate to behaviors like swimming and crawling, though additional experimental support would be beneficial and is rightly considered outside the scope of this paper.

Overall, the revisions strengthened the theoretical basis, improved clarity, and better connected the work to existing research. The authors addressed key concerns and openly acknowledged remaining limitations, enhancing the manuscript’s credibility.

**Have the authors made all data and (if applicable) computational code underlying the findings in their manuscript fully available?**

Reviewer #1: Yes

Reviewer #3: Yes

PLOS authors have the option to publish the peer review history of their article (what does this mean? ). If published, this will include your full peer review and any attached files.

**Do you want your identity to be public for this peer review?** For information about this choice, including consent withdrawal, please see our Privacy Policy .

Reviewer #1: No

Reviewer #3: No

---

## [Editor Report · Acceptance letter]

PCOMPBIOL-D-24-02033R1

The roles of feedback loops in the Caenorhabditis elegans rhythmic forward locomotion

Dear Dr Fang,

I am pleased to inform you that your manuscript has been formally accepted for publication in PLOS Computational Biology. Your manuscript is now with our production department and you will be notified of the publication date in due course.

With kind regards,

Zsofia Freund
